# Synthesis and Processing Parameter Optimization of Nano-Belite via One-Step Combustion Method

**DOI:** 10.3390/ma15144913

**Published:** 2022-07-14

**Authors:** Hongfang Sun, Weixing Lian, Xiaogang Zhang, Wei Liu, Feng Xing, Jie Ren

**Affiliations:** 1Guangdong Provincial Key Laboratory of Durability for Marine Civil Engineering, College of Civil and Transportation Engineering, Shenzhen University, Shenzhen 518060, China; sunhf03@szu.edu.cn (H.S.); 2070474070@email.szu.edu.cn (W.L.); szzxg@szu.edu.cn (X.Z.); liuwei@szu.edu.cn (W.L.); xingf@szu.edu.cn (F.X.); 2Department of Civil, Environment, and Architectural Engineering, University of Colorado Boulder, Boulder, CO 80309, USA

**Keywords:** nano-belite cement, one-step combustion method, parameter optimization, morphology, urea

## Abstract

This paper proposes a new chemical combustion method for the synthesis of nano-low-carbon belite cement via a simple one-step process without using any oxidizers, and related mechanisms are briefly introduced. The starting materials used, including micro-silica (silica fume) as a byproduct of the metallurgic industry and CaCO_3_ powders, are of great abundance, and the processing parameters involved were optimized using a series of systematic experiments based on X-ray diffraction (XRD) and the Rietveld fitting method. Besides, the properties of the synthesized belite cement were characterized by the Brunauer–Emmett–Teller (BET) technique and scanning electron microscopy (SEM). Experimental results revealed that the optimized fuel agent was urea with a dosage of 4.902 times that of the starting materials by mass, and the corresponding holding temperature and time were 1150 °C and 2 h, respectively. In addition, the CaO/(SiO_2_ + CaO) for the starting materials should be set at 62.5% by mass ratio. BET and SEM results showed that the obtained belite cement had a specific surface area of 11.17 m^2^/g and a size of around 500 nm or even smaller in spherical shapes, suggesting that this method was successfully implemented. Thus, it can be a promising approach for the synthesis of nano-belite particles as a low-carbon construction material, which could be used more in the near future, such as for low-carbon concrete productions.

## 1. Introduction

Belite (an impure form of β-Ca_2_SiO_4_) is one of the phase materials while producing traditional ordinary Portland cement (OPC) clinker based on the conventional calcination method. Belite-rich cement has received immense interest not only because of its relatively higher later-age strength development, but also its better durability over a long service life thanks to more formation of C-S-H gels and fewer cracks during the early-age hydration [1,2]. In addition, the production of belite clinker requires much less limestone and energy consumption (1350 kJ/kg), as compared to its counterpart alite (1810 kJ/kg), as the main component in tradition OPC, thus significantly reducing the total CO_2_ emissions [3]. According to previous studies [4,5,6], the total amount of CO_2_ emissions to the atmosphere for the production of belite clinker is only about half per unit mass of traditional OPC clinker. However, the low hydraulic reactivity of belite produced by conventional calcination of calcium carbonate (CaCO_3_) and silicon dioxide (SiO_2_) with a temperature higher than 1200 °C and the resultant low early-age strength significantly limit its wider application. The low reactivity can be due to the intrinsic crystallinity and relatively compact microstructure of belite cement [3,4]. Thus, currently, the production of highly reactive belite has been an increasing concern. Many efforts have been made to synthesize belite with good hydraulic reactivity, such as the sol-gel method, hydrothermal method, flame spray pyrolysis, or the spray-drying method, or simple ways based on solid-state reactions using nano-silica [2,5,6,7,8,9]. Despite the relatively higher hydraulic reactivity of the obtained belite, these methods are either quite complicated or not cost-effective enough for large-scale production because of the costly starting materials and/or long preparation time.

In this regard, preparation of nano-belite using nanotechnology seems to be a promising approach as nanoscale belite particles are assumed to be of high chemical reactivity due to the increased specific surface area compared to their normal size formed in conventional OPC clinker. Besides, the increased specific surface area would have a potential nucleation effect which could also improve the strength and durability of OPC concrete when partial OPC is replaced by nano-belite, similar to that introduced by nano-silica [7,8,9,10], nano-TiO_2_ particles [11,12], aluminum oxide nanoparticles [13], and even nano-cement [14,15,16]. According to [16], nano-cement led to reduced shrinkage and permeability and increased strength, as well as long-term durability of concrete.

With the purpose of preparing nano-belite using a simple, cost-effective method, a combustion reaction seems to be a possible solution as it can produce very fine powders, although nanoscale powders’ production may need further improvements [17]. Different from conventional calcination, the chemical combustion method involves an organic fuel source (also known as a fuel agent) which could not only provide heat required for the chemical reaction, but release a certain amount of gases, leading to a high pressure for the reaction products formed during this process within a relatively limited space [18]. Hence, a rapid and self-sustained reaction is achieved, and the final reaction products are pressurized to fine particles. This process is to some extent similar to the formation of volcanic ash during explosive volcanic eruptions, where spherical pellets of ash are created. During this process, powders are pelletized because of the significant impact from a mixture of various gases which is released out from volcanoes forced by high pressure from the inner part of volcanoes. Similarly, a recent study used this mechanism to pelletize powders by using an automatic press with 7–9 tons of pressure [19]. However, this research pelletized the intermediate powder product first, followed by calcination at elevated temperature, which is a two-step process. Based on the theoretical mechanism, a schematic presentation of the preparation process of fine belite particles is shown in Figure 1. It can be seen that smaller and round-shaped belite cement particles were gradually formed under high pressure during the process of gas release.

Despite the widely known simple mechanisms, preparing nano-belite cement via the combustion method has been poorly documented [20]. To the best of the authors’ knowledge, this study synthesized nanoscale belite via a one-step combustion method using relatively cheap and simple starting materials without any usage of an oxidizer. Moreover, for the first time, the processing parameters were optimized, including the type and relative amount of fuel agent, mass ratio of the starting raw materials, as well as the holding temperature and the time, step-by-step. Furthermore, related basic properties of the prepared belite cement were also examined.

## 2. Experimental Program

### 2.1. Raw Materials

The basic raw materials were ultrafine micro-silica (SiO_2_ ≥ 90.0%) and CaCO_3_ powders (CaCO_3_ ≥ 99.0%), which were provided by Guangzhou Tengri Building Material Co., Ltd. (Guangzhou, China) and Xilong Scientific Co., Ltd. (Shantou, Guangdong, China), respectively. The micro-silica has an average size of 0.1–0.3 μm with a specific surface area of 18–28 m^2^/g. Its specific chemical composition is listed in Table 1. Several potential organic fuel agents used were kerosene, ethanol, urea, rice husk, and charcoal. All these fuel agents were relatively cheap and combustible. Among them, urea (H_2_NCONH_2_ > 99.0%) is a white crystalline powder, with its composition shown in Table 2, which was provided by the same company as the CaCO_3_ powders.

### 2.2. Synthesis of Belite Cement

The combustion method was applied to prepare nano-belite particles, which is described as follows: Firstly, each of the fuel agents in their original form was added into the mixture of micro-silica and CaCO_3_ powders, followed by adding a small amount of water for complementary purposes, and then fully mixed. After that, the whole mixture was immediately placed in an alumina crucible and calcined in a muffle oven at a predetermined temperature (holding temperature) for a certain period of time (holding time). The obtained specimens were then allowed to cool down rapidly in air at room temperature (20 ± 2 °C), roughly at a rate of 60 °C/min, to prevent the polymorphic transformation from β-C_2_S to γ-C_2_S with no hydraulic reactivity, ensuring the stabilization of belite phase with high reactivity. Finally, the specimen was slightly crushed for collection and further closely stored in sealed plastic bags before testing. A schematic presentation of the synthesis process of the belite is shown in Figure 2.

### 2.3. Optimization of Synthesis Parameters

#### 2.3.1. Selection of Fuel Agent

Firstly, five different types of fuel agents were used, namely rice husk, kerosene, ethanol, charcoal, and urea. The reason why these fuel agents were employed is that these are either residues such as rice husk, produced as a byproduct from rice milling, or can be produced on a large-scale with relatively low cost, such as ethanol and urea. A better fuel agent is assumed to result in a higher concentration of reactive Ca_2_SiO_4_ (including α, α′, and β polymorphs), with few residues or impurities in the final products. An increased volume of the final reaction product is preferred as this indicates that enough gas is released, accompanied with high pressure to pelletize the obtained belite particles. For this, the holding temperature and time were kept at 815 °C and 0.5 h, respectively, as it has been confirmed that belite can be obtained with a temperature greater than 815 °C [21,22,23]. This temperature is also the minimum value for decarbonation [24]. It is worth pointing out that before the incineration process, the five fuel agents were calcined alone first without adding any raw materials to examine their resultant residues. A better fuel agent introduces fewer solid residues to the system after incineration as solid residues could negatively influence the purity of the final reaction products. Later, each of the fuel agents were mixed with the raw material (micro-silica and CaCO_3_ powders) and then calcined at 815 °C for 0.5 h. The mass ratio between the fuel agent and raw material was fixed at 4.902 as the basic value based on pilot studies [20]. A basic mix design of the test is presented in Table 3 and all the other changes in the following sections can be calculated based on the detailed introduction.

The appearances of the mixture before and after the sintering were monitored. To quantify and compare different mineralogical phases of the reaction products, X-ray diffraction (XRD) analysis along with the Rietveld refinement method [4,25] were performed using a Bruker D8 instrument with a scanning range from 10° to 70°, using a sampling time of 0.3 s/step and a step size of 0.02° 2θ. The operation voltage and current were 40 V and 40 mA, respectively. Jade 9.0 software with the PDF-2 database was used for identifying different phases involved in the production of belite. The components used for refinement were: SiO_2_, CaO, Ca_2_(SiO_4_), Ca_3_SiO_5_, and Ca_3_(SiO_3_)_3_/CaSiO_3_. After selecting the proper fuel agent, other parameters, i.e., the CaO/SiO_2_ ratios, holding time, and holding temperature, as well as the relative content of the fuel agent, were subsequently optimized.

#### 2.3.2. Determination of CaO/(SiO_2_ + CaO) Ratios

According to the main reaction equations as described below based on the combustion method, stoichiometrically, the mass ratio of CaO/(SiO_2_ + CaO) is 65.12% when the reactants (CaCO_3_ powders and micro-silica fume) achieve the maximum degree without any surplus. However, because of the impurities contained in the corresponding reactants and other influences, a 100% degree of chemical reaction between the two reactants is almost impossible. Considering the lower purity of micro-silica fume (SiO_2_ ≥ 90.0%) compared to that of the CaCO_3_ powders (CaCO_3_ ≥ 99.0%), the SiO_2_ is in short supply, and thus the CaO produced after the calcination of CaCO_3_ powders should be slightly lower. As a result, the apparent CaO/(SiO_2_ + CaO) ratio should be slightly smaller than 65.12%. Thus, in this study, five different CaO/(SiO_2_ + CaO) ratios were employed, which were 65%, 62.5%, 60%, 57.5%, and 55%.
$${\mathrm{CaCO}}_{3}\to \mathrm{CaO}+{\mathrm{CO}}_{2}$$
$$2\mathrm{CaO}+{\mathrm{SiO}}_{2}\to 2\mathrm{CaO}\cdot {\mathrm{SiO}}_{2}$$

Similar to the selection of the fuel agent, a higher concentration of Ca_2_SiO_4_ in the final mixture of products implies a better CaO/(SiO_2_ + CaO) ratio. Besides, a smaller amount of residual SiO_2_ is also preferable since micro-silica fume is relatively more expensive compared to CaCO_3_ powders. At this time, the fuel agent was the one determined from the first section, with an adjusted holding time at 2 h and a holding temperature at 1050 °C, which will be discussed in detail later.

#### 2.3.3. Optimization of Holding Time

Based on the optimized fuel agent and CaO/(SiO_2_ + CaO) ratio, the effects of different holding times on the final calcination products were investigated. Specifically, 0.5, 1, 2, 3, and 4 h of holding time were used, and the mineralogical compositions as well as their relative content in the corresponding reaction products under each condition were analyzed by XRD analysis. The holding temperature this time was 1050 °C for all conditions.

#### 2.3.4. Optimization of Holding Temperature

After the determination of the holding time, different holding temperatures were attempted, i.e., 850, 950, 1000, and 1150 °C. The reason for the highest temperature of 1150 °C used here is that belite cement can be obtained at this temperature [26], although a higher one might be required for certain conditions [27]. In addition, the maximum limit temperature value for the muffle furnace is also 1150 °C. A lower temperature is usually preferred because of the lower energy consumption involved.

#### 2.3.5. Optimization of Fuel Agent Content

Based on the above results, five contents of fuel agent were used, which were 80%, 90%, 100%, 110%, and 120% of that of the original amount of fuel agent used by mass (basic level), which was about 4.902 times the total mass of micro-silica and CaCO_3_ powders.

### 2.4. Characterization of Belite Cement

The specific surface area of the prepared belite nano-cement was measured with a Tri-Star II Surface Area and Porosity Analyzer based on the Brunauer–Emmett–Teller (BET) method. Besides, its original morphology was observed using a scanning electron microscope (Thermo APREO S(A5-112), The Netherlands) in secondary electron mode. After preparation, small belite cement particles were dried in a vacuum oven at 50 °C first to eliminate any hydration effect due to the water ‘contamination’, then placed on conductive double-sided adhesive carbon tapes to provide a high-conductivity surface layer prior to the analysis [20].

## 3. Results and Discussion

### 3.1. Optimization of Synthesis Parameters

In general, a higher content of Ca_2_SiO_4_ in total, α′-Ca_2_(SiO_4_), and β-Ca_2_SiO_4_ with high reactivity [28] fabricated at lower temperatures and with less holding time are preferred. In addition, obtaining final products with fewer residual raw materials is also a key principle to ensure the high purity of products when selecting optimal parameters.

#### 3.1.1. Determination of Fuel Agent

The appearances of the fuel agents before calcination and corresponding calcined residues are shown in Figure 3. It can be seen that there are no obvious residues for kerosene, ethanol, and urea since the main reaction products are gases such as water vapor, CO_2_, and NH_3_. In comparison, some rice husk ash and charcoal ash were observed. Rice husk ash is rich in amorphous silica with a highly microporous cellular structure [29,30], which may have an adverse effect on the final products as the highly microporous structure could absorb a lot of water, impeding a further reaction of belite. The main elements in charcoal ash are Ca, Mg, K, Na, and P [31], thus adding more irrelevant elements into the final reaction product system. Therefore, it is apparent that kerosene, ethanol, and urea are better compared to rich husk and charcoal to be used as the fuel agent.

The mixture of each fuel agent and raw material before and after calcination is shown in Figure 4a,b. For the mixtures before calcination (shown in Figure 4a), it is apparent that the combined micro-silica and CaCO_3_ powders were not well-dispersed in the liquid kerosene, most of which just precipitated and accumulated on the bottom of the crucible. However, the micro-silica and CaCO_3_ powders were distributed in ethanol solution homogenously, which is considered better than the kerosene. The worse dispersion state of micro-silica and CaCO_3_ powders in the kerosene might be associated with its oily characteristic. A slurry was formed when mixing the urea with the micro-silica and CaCO_3_ powders. The color of the mixtures with these three fuel agents is similar to the original combination of micro-silica and CaCO_3_ powders. For the raw materials mixed with rice husk or charcoal, the obtained mixture showed no color of the raw materials, and it seems that the rice husk or charcoal dominates the whole mixture. This can be explained by considering the relatively larger volume of the rice husk or charcoal because of their lower density than that of the raw materials. Thus, these results suggest that kerosene, rice husk, and charcoal are not suitable to be used as fuel agents because they are not able to achieve a homogeneous mixture with the raw materials.

After calcination in the muffle, as shown in Figure 4b, it was found that almost all the reaction products for kerosene and ethanol mainly accumulated under the bottom of the crucible with a color of light grey. Besides, the products seemed to be very fragile and crispy with some obvious rifts. The fragile and easily damaged products are associated with the gas released during the calcination process, which indirectly suggests that this method is feasible. The reaction products formed for urea were relatively fluffy near the crucible mouth, indicating a significant increase in the total volume. Besides, the color was snow white, which suggests a relatively high purity. There were a lot of burned grey residues and still burning black charcoals for rice husk and charcoal used as the fuel agent, respectively. Based on these observations, urea is assumed to be a better fuel agent, since the increased volume of the reaction products indicates a large amount of gases released during reaction and the pure white color demonstrates a high content of belite in the final products.

To further verify the assumption, the mineralogical composition of the obtained reaction products and corresponding content determined by using semi-quantitative XRD analysis is shown in Figure 5 and Table 4. It shows that the highest percentage of Ca_2_SiO_4_ in total and β-Ca_2_SiO_4_ was obtained when urea was used as the fuel agent, which was 55.3% and 32.7%, respectively. Kerosene and ethanol led to similar productivity of Ca_2_SiO_4_, the latter of which had a slightly higher content, probably due to the good dispersion of raw materials in ethanol compared to that in the kerosene. Not surprisingly, the charcoal and rice husk resulted in the lowest content of Ca_2_SiO_4_ in total and β-Ca_2_SiO_4_, less than 30% and just at 20%, accordingly. All these results were consistent with the aforementioned appearances of the mixture before and after calcination. It was also noticed that the percentage of SiO_2_ present in the final reaction products was relatively low for urea (3.4%) and ethanol (3.0%), suggesting a higher reaction degree compared to other fuel agents, except for rice husk, since most of the residual SiO_2_ might be introduced by the rice husk. It is probably because the original micro-silica and CaCO_3_ powders had a rather complete reaction accompanied by the combustion of rice husk and the formation of rice husk ash with amorphous aluminosilicate rich in SiO_2_. The highest percentage of CaSiO_3_ (a reaction product formed when the heating temperature is not high enough) seen in the charcoal-added mixture can be due to its continuous burning, which requires a lot of heating energy, leading to a lower temperature for calcination and thus an incomplete reaction.

In conclusion, urea was hereafter selected as the optimized fuel agent not only because of the much greater volume of the reaction products, but also the highest percentage of Ca_2_SiO_4_ phases obtained with fewer impurities, including Ca_3_SiO_5_ and SiO_2_. It is also worth noting that one more advantage for urea is its relatively low cost, which makes it the most frequently used fuel among other organic fuels [17]. It is noteworthy that the total concentration of Ca_2_SiO_4_ was not high enough, which is why for the following tests the holding temperature and time were set at 1050 °C and 2 h, respectively [32,33].

#### 3.1.2. Determination of CaO/(SiO_2_ + CaO) Ratios

After the selection of urea as the fuel agent, percentages of various mineralogical phases formed under different ratios of CaO/(SiO_2_ + CaO), namely 65%, 62.5%, 60%, 57.5%, and 55%, were calculated based on the XRD semi-quantitative analysis. The corresponding patterns and the final results are shown in Figure 6 and Table 5, respectively. The highest Ca_2_SiO_4_ in total and the second highest value of β-Ca_2_SiO_4_ with relatively high hydration reactivity were achieved when the ratio was 62.5%, which were 79.5% and 69.9%, respectively. The highest content of another belite polymorph with high reactivity, α′-Ca_2_(SiO_4_), was 2.5% with CaO/(SiO_2_ + CaO) at 65%, followed by 2.4% when CaO/(SiO_2_ + CaO) was 62.5%. Besides, the percentage ratio of the residual SiO_2_ considering CaO/(SiO_2_ + CaO) at 62.5% was 3.1%, the second lowest one among the five mixtures of reaction products, after that of the CaO/(SiO_2_ + CaO) ratio at 65%. In addition, there was no significant difference in the content of CaO for all systems, which ranged between 0.2% and 0.5%, indicating that the CaO from CaCO_3_ powders almost completely reacted with the micro-silica powders. As the aim of the research was to obtain highly reactive belite cement with higher purity, the 62.5% ratio of CaO/(SiO_2_ + CaO) was determined as the optimal ratio.

#### 3.1.3. Determination of Holding Time

According to the selected fuel agent and the CaO/(SiO_2_ + CaO) ratio at 62.5%, different holding times (0.5, 1.0, 2.0, 3.0, and 4.0 h) were used to examine the effect of holding time on the final reaction products and to determine the most suitable holding time. Based on the XRD patterns (shown in Figure 7), calculated percentages of various mineralogical phases in the final products are listed in Table 6.

According to the results, the total content of Ca_2_SiO_4_ increased first from 74.2% for 0.5 h up to 87.2% with the holding time at 2.0 h, and then decreased slightly to 86.1% and 86.4%, respectively, when the holding time was 3.0 and 4.0 h. This outcome demonstrates that a longer holding time did not necessarily lead to higher content of Ca_2_SiO_4_, and there is an optimal holding duration above which no more Ca_2_SiO_4_ could be formed. This might be associated with the limited temperature, in which case a further reaction duration had no significant effect on increasing the amounts of the final products. The content of β-Ca_2_(SiO_4_) decreased first from 56.0% (0.5 h) to only 28.1% (1.0 h), then increased up to 40.7% (2.0 h) and 59.4% (3.0 h), followed by a sharp decrease down to only about 31.4% when the holding time was 4.0 h. A 2.0 h holding time led to the highest content of α′-Ca_2_(SiO_4_) (8%), with the highest hydraulic reactivity [34]. Besides, it also can be seen that CaO and SiO_2_ content experienced no significant reductions after 2.0 h of suspension, indicating that a longer holding time would not facilitate further reactions, which might also be associated with the holding temperature selected in this study. Considering that a reduced holding time usually leads to less energy consumption, a 2 h holding time was chosen for the following tests.

#### 3.1.4. Determination of Holding Temperature

XRD patterns of the reaction products prepared using different holding temperatures are shown in Figure 8 and corresponding semi-quantitative results are tabulated in Table 7. It is obvious that the overall content of Ca_2_SiO_4_ increased as the temperature increased from 815 to 1150 °C, from 57.4% to 87.2%. At the same time, the content of unreacted CaO and SiO_2_ decreased gradually, indicating a continuous reaction process. As a higher yield of belite in total is preferred, thus 1150 °C was considered as the optimal temperature. This belite content (87.2%) was relatively high. Besides, the content of another two highly reactive polymorphs, β-Ca_2_SiO_4_ and α′-Ca_2_(SiO_4_), was relatively high (40.7% and 8.0%, respectively) behind the highest value, which is also acceptable.

#### 3.1.5. Determination of Fuel Agent Content

Finally, based on the predetermined optimized process parameters, the optimal amount of fuel agent was determined. As mentioned earlier, the original amount of urea used was 4.902 times that of the combined mass of micro-silica and CaCO_3_ powders. XRD patterns and corresponding calculated percentage ratios of various products are presented in Figure 9 and Table 8, respectively. It is clear that the content of Ca_2_SiO_4_ in total increased first from 70.9% to 87.2% when the amount of urea grew from 80% to 100%, followed by a small reduction, 81.8% (110%) and 73.2% (120%). The decreased amount of Ca_2_SiO_4_ in total can be attributed to the fact that an excessive amount of urea would require more O_2_, which may then result in an inadequate combustion. As a result, the actual temperature for calcination might not be high enough to achieve a high production of Ca_2_SiO_4_. The minimum percentage ratio of CaO and SiO_2_ can be seen in the reaction products for 110% and 100% of urea, which was 5.8% and 3.1%, respectively. Thus, since 100% of urea led to the highest content of Ca_2_SiO_4_ in total and the lowest percentage ratio of unreacted SiO_2_, 100% of urea was finally selected.

After this process, all the process parameters involved in the synthesis of nano-belite cement were optimized, which are summarized in Table 9.

### 3.2. Properties of Belite Cement

Based on Figure 10, it can be seen that the obtained belite presents loosely aggregated clinker of small, spherical particles with homogenous morphology, with a size of about 500 nm or less. According to the research [35], this size can be used for concrete. Then, the clinker was slightly ground to obtain the finished belite cement. This grinding process uses much less energy than that of the traditional OPC due in large part to its fine and porous nature [36]. According to the BET result, this type of belite had a specific surface area of 11.17 m^2^/g, which was much greater than that of a type P·O 42.5 OPC (provided by Qufu Zhonglian Cement Co., Ltd., Qufu, China) with a surface area of 2.57 m^2^/g. This high surface area was even about 2–3 times larger compared to the belite cement (3.0 m^2^/g) prepared using calcite and silica fume, as reported in another study [17], or other belite synthesized via more complex methods [19,37]. Thus, this result reveals that the one-step synthesis of nano-belite cement based on the chemical combustion of solid-state raw materials using urea as the fuel agent is quite successful. Its nanoscale particle size would not only enhance its own hydraulic reactivity but would also be beneficial for overall strength improvement due to nucleation and pore-filling effects. Further research will be conducted to shed light on these effects.

## 4. Conclusions

In this research, nano-belite cement was manufactured for the first time using a benign chemical combustion approach via a simple one-step process. Micro-silica and CaCO_3_ powders were used as the raw materials and urea was selected as the best fuel agent among five potential candidates. Besides, related process parameters, including CaO/(SiO_2_ + CaO) ratios, holding temperature, holding time, and dosages of fuel agent, were optimized step-by-step based on a series of systematic experiments. As a result, the corresponding optimized values were 62.5%, 1150 °C, 2 h, and 4.902 times that of the total mass of micro-silica and CaCO_3_ powders. Moreover, the prepared belite had a size of no more than 500 nm and a specific surface area of up to 11.17 m^2^/g, the latter of which was much greater than the belite particles fabricated in previous literature, suggesting that this one-step combustion method is promising. Considering that the micro-silica and CaCO_3_ powders as well as the urea used are relatively cheap and environmentally friendly with no introduction of any pollutants, this method to some extent paves a way for large-scale fabrication of nano-belite cement, which can be further used in massive concrete production. In addition, this environmentally benign and simple method is expected to be used for the synthesis of other nanoscale particles with relatively low energy consumption.

## Figures and Tables

**Figure 1 materials-15-04913-f001:**
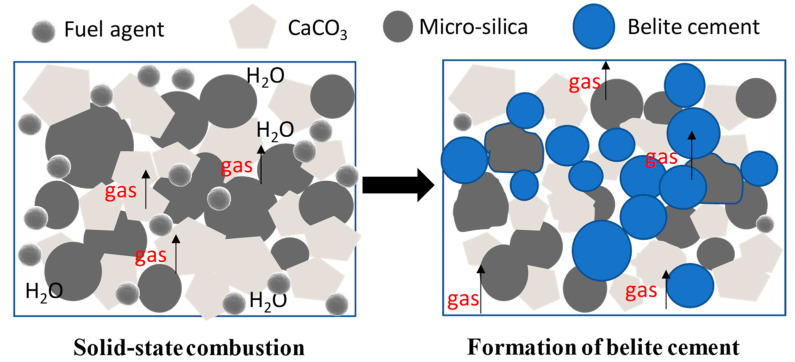
Schematic presentation of the formation of belite via a one-step combustion method.

**Figure 2 materials-15-04913-f002:**
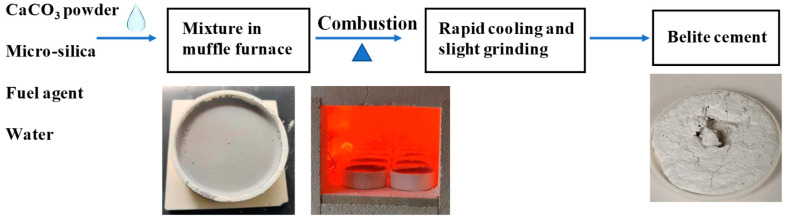
A flow chart showing the one-step synthesis process of the belite cement.

**Figure 3 materials-15-04913-f003:**
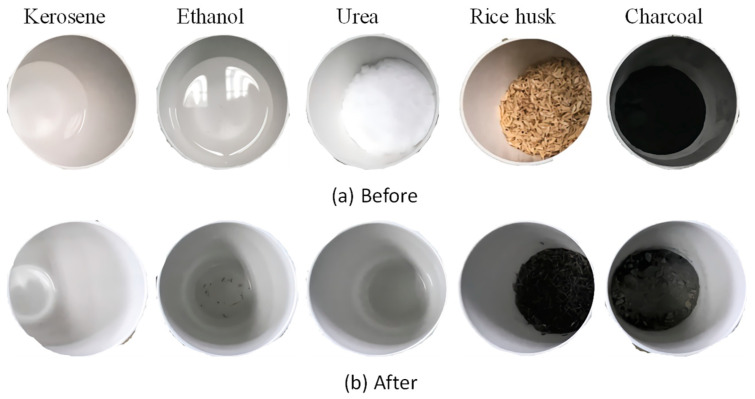
Appearances of the five fuel agents before and after calcination.

**Figure 4 materials-15-04913-f004:**
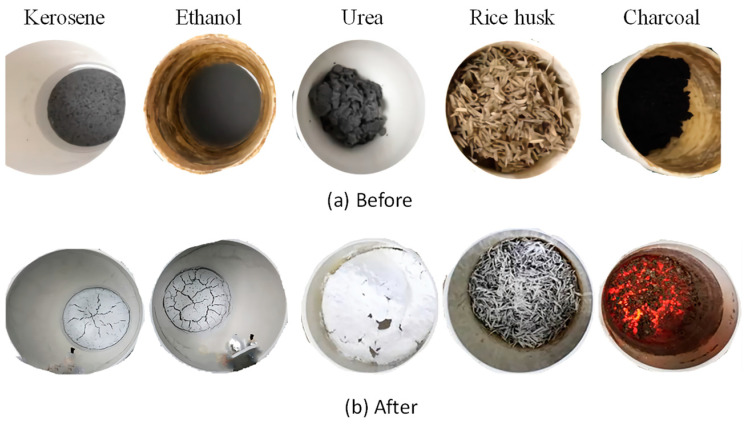
Appearances of the mixed fuel agent and raw materials (micro-silica and CaCO_3_ powders) before and after calcination.

**Figure 5 materials-15-04913-f005:**
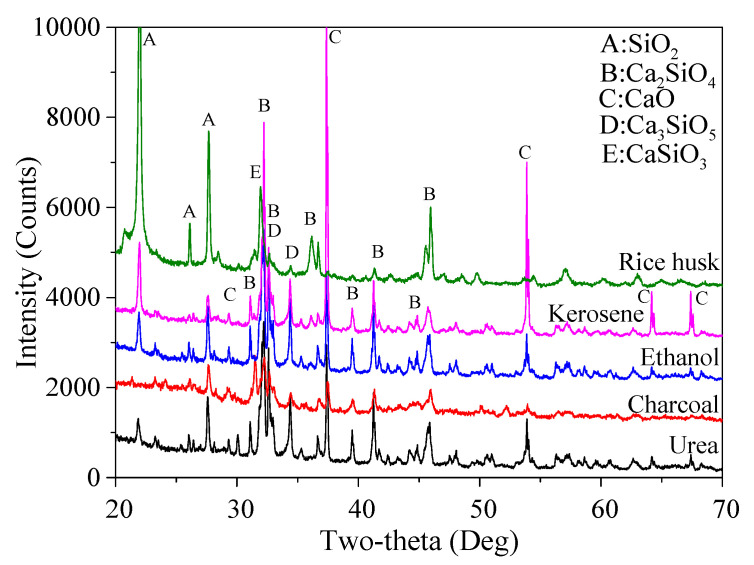
XRD patterns of the reaction products using five types of fuel agents.

**Figure 6 materials-15-04913-f006:**
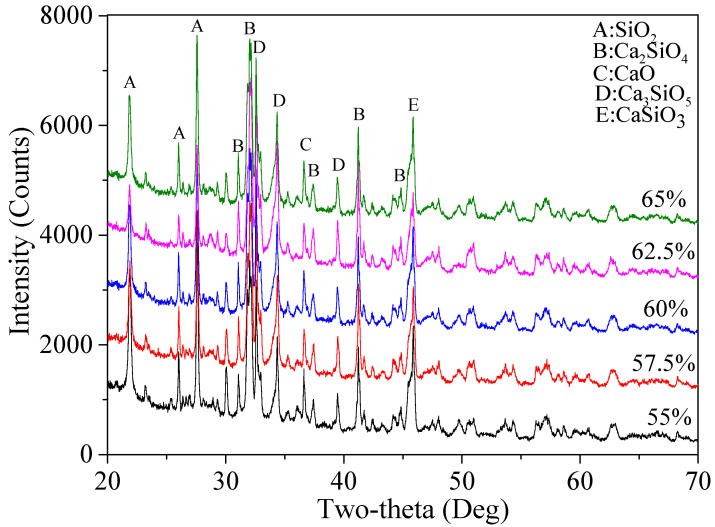
XRD patterns of the reaction products using five different CaO/(SiO_2_ + CaO) ratios.

**Figure 7 materials-15-04913-f007:**
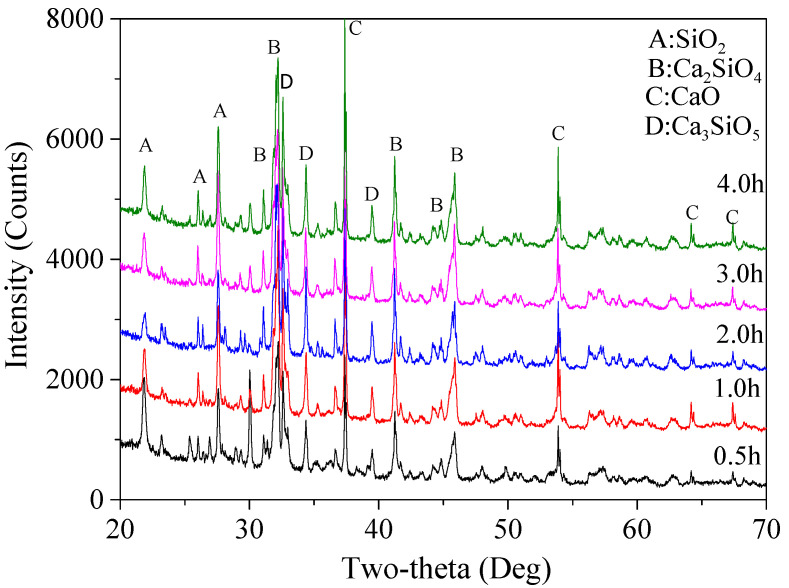
XRD patterns of the reaction products using five different holding times.

**Figure 8 materials-15-04913-f008:**
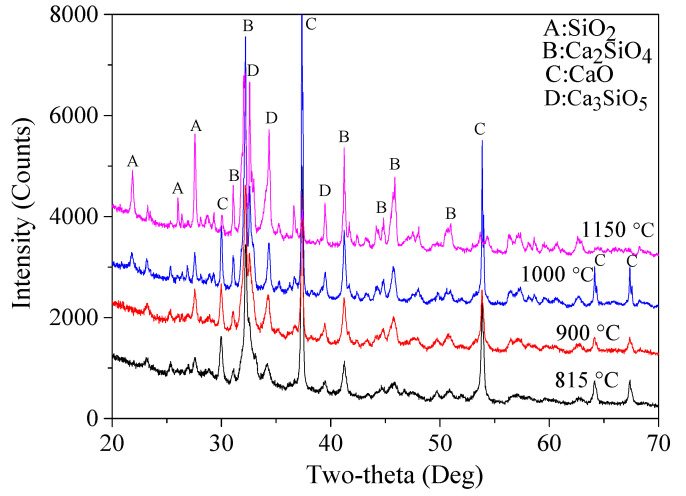
XRD patterns of the reaction products using four different holding temperatures.

**Figure 9 materials-15-04913-f009:**
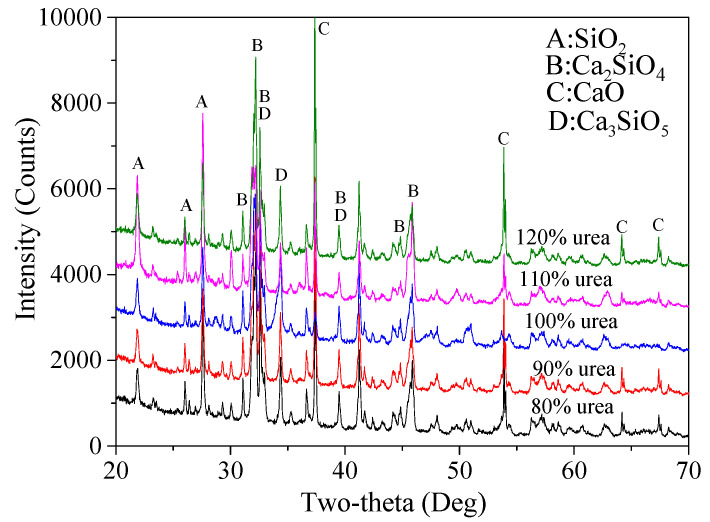
XRD patterns of the reaction products using the five different dosages of urea.

**Figure 10 materials-15-04913-f010:**
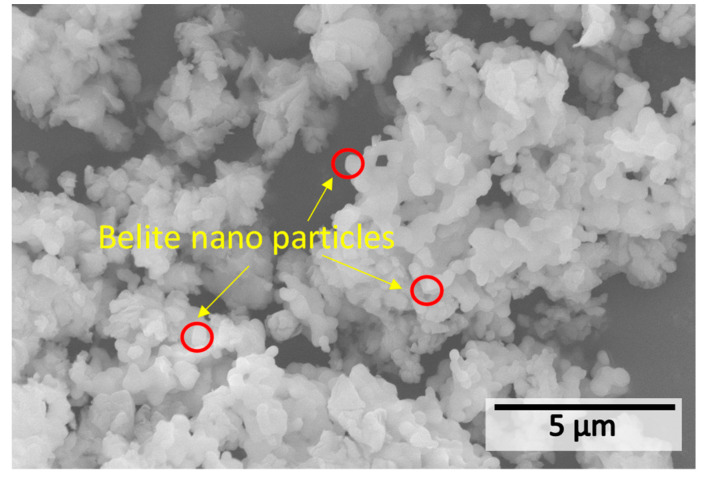
SEM image of the nano-belite cement after the preparation process.

**Table 1 materials-15-04913-t001:** Chemical composition of the micro-silica used in this study.

Oxides	SiO_2_	Fe_2_O_3_	Al_2_O_3_	CaO	MgO	K_2_O	Na_2_O	LOI
Contents (%)	95.55	0.41	0.32	0.19	0.30	0.50	0.21	2.68

LOI: Loss on ignition at 1000 °C.

**Table 2 materials-15-04913-t002:** Composition of the urea used in this study.

Components	CO(NH_2_)_2_	IM	Chlorine	Sulfate	NH_3_	Fe	HM	Biuret	LOI
Contents (%)	99.0	0.005	0.003	0.001	0.005	0.0002	0.0002	0.2	0.01

IM: insoluble matter, HM: heavy metal.

**Table 3 materials-15-04913-t003:** Basic mix design for selecting the optimal fuel agent.

Foaming Agent	CaCO_3_	Micro-Silica	Deionized Water	Temperature (°C)	Holding Time (h)
Rice husk (14.17 g)	2.16 g	0.73 g	5 g	815	0.5
Urea (28.38 g)	4.32 g	1.47 g	5 g	815	0.5
Charcoal (28.38 g)	4.32 g	1.47 g	9 g	815	0.5
Ethanol (28.38 g)	4.32 g	1.47 g	0	815	0.5
Kerosene (28.38 g)	4.32 g	1.47 g	0	815	0.5

**Table 4 materials-15-04913-t004:** Percentage mass ratio (%) of mineralogical compositions of the final products for different fuel agents.

	Fuel Agent	Urea	Charcoal	Kerosene	Ethanol	Rice Husk
Products						
β-Ca_2_(SiO_4_)		**32.7**	18.0	26.9	29.4	17.2
α′-Ca_2_(SiO_4_)		**1.2**	0.6	0.4	0.7	0.2
α-Ca_2_(SiO_4_)		18.4	10.2	12.6	14.2	2.6
Ca_2_(SiO_4_) in total		**55.3**	28.2	40.5	44.3	20.0
Ca_3_SiO_5_		4.0	1.4	3.6	4.6	1.0
CaO		19.6	5.9	27.5	9.0	5.1
Ca_3_(Si_3_O_9_)		17.4	60.4	23.2	39.0	40.6
SiO_2_		3.4	4.1	5.3	3.0	33.3

The highest values for certain key products are highlighted in bold font.

**Table 5 materials-15-04913-t005:** Percentage mass ratio (%) of mineralogical compositions of the final products for five different CaO/(SiO_2_ + CaO) ratios.

	CaO/ (SiO_2_ + CaO)	55%	57.5%	60%	62.5%	65%
Products						
β-Ca_2_(SiO_4_)		**72.1**	60.2	68.1	69.9	57.1
α′-Ca_2_(SiO_4_)		0.2	0.3	1.3	2.4	**2.5**
α-Ca_2_(SiO_4_)		4.4	16.6	8.9	7.2	19.3
Ca_2_(SiO_4_) in total		76.7	77.1	78.3	**79.5**	78.9
Ca_3_SiO_5_		0.7	2.9	3.0	2.7	5.2
CaO		0.2	0.3	0.3	0.3	0.5
Ca_3_(SiO_3_)_3_		17.3	14.8	14.8	14.4	13.0
SiO_2_		5.1	4.9	3.6	3.1	2.4

The highest values for certain key products are highlighted in bold font.

**Table 6 materials-15-04913-t006:** Percentage mass ratio (%) of mineralogical compositions of the final products for five different holding times.

	Holding Time	0.5 h	1.0 h	2.0 h	3.0 h	4.0 h
Products						
β-Ca_2_(SiO_4_)		56.0	28.1	40.7	**59.4**	31.4
α′-Ca_2_(SiO_4_)		6.3	6.4	**8.0**	0.6	5.6
α-Ca_2_(SiO_4_)		11.9	46.9	38.5	20.1	43.4
Ca_2_(SiO_4_) in total		74.2	81.4	**87.2**	86.1	86.4
Ca_3_SiO_5_		1.1	0.3	0.2	0.5	0.4
CaO		6.3	12.0	9.3	6.7	6.9
CaSiO_3_		4.4	1.5	0.2	0.4	3.2
SiO_2_		14.0	4.8	3.1	6.1	3.1

The highest values for certain key products are highlighted in bold font.

**Table 7 materials-15-04913-t007:** Percentage mass ratio (%) of mineralogical compositions of the final products for four different holding temperature values.

	Holding Temperature	815 °C	900 °C	1000 °C	1150 °C
Products					
β-Ca_2_(SiO_4_)		35.3	38.0	**47.3**	40.7
α′-Ca_2_(SiO_4_)		**11.7**	0.7	2.2	8.0
α-Ca_2_(SiO_4_)		10.4	31.9	27.3	38.5
Ca_2_(SiO_4_) in total		57.4	70.6	76.8	**87.2**
Ca_3_SiO_5_		0.2	0.3	0.6	0.2
CaO		23.3	13.5	12.3	9.3
Ca_3_(SiO_3_)_3_		7.3	8.1	4.5	0.2
SiO_2_		11.8	7.5	5.8	3.1

The highest values for certain key products are highlighted in bold font.

**Table 8 materials-15-04913-t008:** Percentage mass ratio (%) of mineralogical compositions of the final products for five different dosage levels of urea.

	Urea Dosage	80%	90%	100%	110%	120%
Products						
β-Ca_2_(SiO_4_)		44.8	47.3	40.7	41.5	53.7
α′-Ca_2_(SiO_4_)		8.8	8.0	8.0	11.4	3.3
α-Ca_2_(SiO_4_)		17.3	17.1	38.5	28.9	16.2
Ca_2_(SiO_4_) in total		70.9	72.4	**87.2**	81.8	73.2
Ca_3_SiO_5_		0.1	0.5	0.2	0.1	0.5
CaO		7.2	9.0	9.3	5.8	10.8
Ca_3_(SiO_3_)_3_		12.6	8.8	0.2	5.7	8.6
SiO_2_		9.2	9.3	3.1	6.6	6.9

The highest values for certain key products are highlighted in bold font.

**Table 9 materials-15-04913-t009:** Final optimized processing parameters for the nano-belite preparation.

Parameters	CaO/(SiO_2_ + CaO)	Fuel Agent	Holding Temperature	Holding Time	Urea/Raw Materials by Mass
Optimized value/item	62.5%	Urea	1150 °C	2 h	4.902

## Data Availability

Data is contained within the article.

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
