# Peer review of "Synthesis and Processing Parameter Optimization of Nano-Belite via One-Step Combustion Method"

_materials, 2022, doi:10.3390/ma15144913_

Round 1
Reviewer 1 Report
This is a good paper working on the low-carbon cement (belite), providing the new insights on the synthesis of it at the lab scale. The topic is meaningful, although it may be not that new. The nano-scale belite could potentially be used as the low carbon cement if the strength requirement is not high. Therefore, the paper is recommended to be published after addressing the following comments:
1. The final sentence of your abstract: it is too far from 'widely used', could the authors please revise it?
2. Also the 'relative cheap': firstly should be 'relatively cheap'; then, the micro silica may be not that cheap. Before the full estimation of your cost, could the authors please remove the relative expressions?
3. Could the authors add the rising temperature curve during the synthesis?
4. Could the authors add some more information about QXRD. May be present in the SI?
5. How about the impurity? If the CaO and SiO2 still present in the synthesized Belite?
Reviewer 2 Report
The paper submitted for review is in the form of a laboratory test report. The paper itself, however, has some shortcomings that need to be supplemented in order to improve its content. There is no information in the paper on carrying out the same type of tests for classic Portland cement or metallurgical cement. Moreover, there is no information on how concrete properties will change when using this type of cement. How will the use of this type of cement affect the frost resistance of concrete? How will the application of the new cement proposal affect the abrasion resistance of concrete? Is the developed new type of cement suitable for use with high-strength concretes? What type of aggregate is recommended for use in the case of using the cement that is the subject of this article? How does the cement used affect the water content of the concrete mix? The paper lacks even a simplified economic analysis - there is no information on the cost of cement production and no assessment of the possibility of its production on an industrial scale. The Reviewer supports the idea of ​​publishing the article after supplementing the explanations. If at the current stage of research it is not possible to answer the above questions, please indicate it.
